# Magneto-Tactile Sensor Based on a Commercial Polyurethane Sponge

**DOI:** 10.3390/nano12183231

**Published:** 2022-09-18

**Authors:** Ioan Bica, Gabriela-Eugenia Iacobescu, Larisa-Marina-Elisabeth Chirigiu

**Affiliations:** 1Advanced Environmental Research Institute, West University of Timisoara, Bulevardul Vasile Pârvan 4, Nr. 4, 300223 Timisoara, Romania; 2Department of Physics, University of Craiova, Strada Alexandru Ioan Cuza, Nr. 13, 200585 Craiova, Romania; 3Department of Pharmacy, “Constantin Brâncuși” University of Târgu-Jiu, Strada Tineretului, Nr. 4, 210185 Târgu Jiu, Romania

**Keywords:** sensor, polyurethane sponge, carbonyl iron, electrical capacity, magnetic field

## Abstract

In this paper, we present the procedure for fabricating a new magneto-tactile sensor (MTS) based on a low-cost commercial polyurethane sponge, including the experimental test configuration, the experimental process, and a description of the mechanisms that lead to obtaining the MTS and its characteristics. It is shown that by using a polyurethane sponge, microparticles of carbonyl iron, ethanol, and copper foil with electroconductive adhesive, we can obtain a high-performance and low-cost MTS. With the experimental assembly described in this paper, the variation in time of the electrical capacity of the MTS was measured in the presence of a deforming force field, a magnetic field, and a magnetic field superimposed over a deformation field. It is shown that, by using an external magnetic field, the sensitivity of the MTS can be increased. Using the magnetic dipole model and linear elasticity approximation, the qualitative mechanisms leading to the reported results are described in detail.

## 1. Introduction

In recent years [1,2,3,4,5], portable and flexible tactile sensors have attracted the attention of the academic community due to their applications, which include measuring physiological parameters in medicine [6,7,8], the robotics industry [9], human–computer interactions [10,11,12], and the creation of portable devices [13,14,15,16]. In general, tactile sensors mimic the human perception of pressure and have the ability to detect the shapes and slipping conditions of objects they come in contact with. As compared with silicon-based devices, flexible materials are more suitable for tactile applications due to their good adhesion, average tensile values, and flexibility [17,18,19,20,21]. Flexible materials such as polyethylene [22], polydimethylsiloxane [23,24], polyurethane [25], and polyimide [26] have been used for manufacturing tactile sensors. Depending on the response function of the sensors, piezoresistive touch sensors [27,28,29], capacitive touch sensors [30], piezoelectric touch sensors [31,32], and optical touch sensors [33] have been developed. To obtain these sensors, metal particles [10], carbon nanotubes (CNT) [34,35], carbon black [36], graphene [37,38,39], or nanowires [22,40] have been introduced into the polymer used in the sensor. In a recent paper [41], a magnetoresistive sensor was fabricated with permanent magnets in the form of Fe-Ga wires and Hall sensors to detect static and dynamic force fields and the degree of rigidity of an object touched. It was a piezoresistive sensor with a sensitivity of 166 mV/m, which was able to detect forces with values up to 3 N and had applications in making prostheses or robots with a precise grip and an intelligent control of objects touched. The development of electrical devices whose response functions are sensitized by the use of an external magnetic field was reported in [42,43,44,45]. In [42,43], the fabrication of mechanical deformation sensors using cotton fiber fabrics with iron carbonyl microparticles and barium titanate nanoparticles was reported. It was shown that the electrical response (resistive, capacitive, and piezoelectric) of electrical devices was stable in time and adjustable in a magnetic field for certain values of hydrostatic pressure. In [44], the magnetic and dielectric effects induced by the magnetic field in a new composite fabric made of cotton fibers and carbonyl iron (CI) microparticles was reported. In [45], the authors reported on the fabrication of a magnetic composite (MS) consisting of a cylindrical polyurethane sponge in which CI microparticles were electrostatically assembled. By applying a static magnetic field, with gradients of up to $1800\;\mathrm{kA}/m^{2}$, superimposed on a medium frequency electric field (f = 1 kHz), a relevant magnetodielectric response was reported. However, the response to repetitive mechanical stress could not be obtained in MS. The manufacturing of magneto-tactile sensors, using the process from [45], is difficult. For this reason, in this paper, we use a commercial low-cost polyurethane sponge with dimensions of $20\times 25\times 5\;{\mathrm{mm}}^{3}$ and a biphasic liquid solution consisting of ethyl alcohol (99%) and CI microparticles to obtain a magnetizable polyurethane sponge (MPS). To fabricate an electrical device, the magneto-tactile sensor (MTS), MPS, and copper foil electrodes are used. The copper foil, 0.50 mm thick and 20 mm wide, has electroconductive adhesive on one side. The adhesive ensures the mechanical-electrical contact between the copper foil and the MPS composite. The purpose of this study is to investigate the response of MTS to compressions exerted on its surface in the absence and presence of the magnetic field. With an RLC bridge, data are obtained regarding the electrical response of the MTS, as well as the response to mechanical and magnetic excitations. On the basis of the results, we conclude that the MTS has a good electrical response for low values of repetitive compressive forces. Moreover, the electrical response of the MTS, for the same mechanical deformations, increases when using a magnetic field.

## 2. Materials and Methods

### 2.1. Materials

The materials used for fabricating the MTS have the technical characteristics provided by the manufacturing companies. The morphology of the materials and the analysis of the chemical elements were performed by scanning electron microscopy (SEM) using an Inspect S PANalytical system (Malvern Panalytical, Malvern, UK) coupled with an energy dispersive X-ray analysis detector (EDX), as follows:

(a)Carbonyl iron microparticles (Sigma-Aldrich, St. Louis, MO, USA) have an average diameter of $d_{m}=5\;\mu m$ and an iron content of at least 97%. The mass density of the CI microparticles is $\rho_{CI}=7.86\;g/cm^{3}$. The particles visualized using SEM were spheres and were confirmed to have an average diameter of 5 μm (Figure 1a). They have a high degree of purity, as shown in Figure 1b.

The magnetization slope of the CI microparticles is shown in Figure 2. It was plotted using the experimental set-up described in [46].

Figure 2 shows a linear dependence of the relative magnetization of the CI microparticles with the intensity of the magnetic field. For magnetic field intensities of H≥540 kA/m, the relative saturation magnetization of the CI microparticles is $\sigma_{\mathrm{SCI}}=195\;{\mathrm{Am}}^{2}/\mathrm{kg}$.

(b)Super absorbent cloth (AC), of the type “Scotch-Brite” (3M, Saint Paul, MN, USA) (20×18×0.5 cm), is made in Italy and contains (see officedirect.ro) 48% viscose, 12% polyester, 25% polyurethane foam, and 15% latex. A piece with the dimensions $20\times 25\times 5\;mm^{3}$ is cut from AC (Figure 3a). By exfoliating a face, the polyurethane sponge (PS) from Figure 3b is obtained, with the dimensions 20×25×4 mm.

The absorbent PS from Figure 3b consists of cells and microfibers, as revealed by the SEM analysis (Figure 4a). Fine and ultrafine particles can be stored in the PS cells. The PS microfibers (Figure 4b) contain carbon and oxygen atoms. The Al atoms are due to the support plate.

(c)Laboratory Reactive Ethyl Alcohol (EA) was purchased from MedAz.ro and has an alcohol content of 96.0%.(d)Copper foil with electroconductive adhesive (CS) (Huizhou Yunze Electronic Technology, Huizhou, China) has a length of 20 m, a width of 3 cm, and a thickness of 0.05 mm. It was acquired from Frugo (Hong Kong, China).

### 2.2. Procedure for Making the MPS and Designing the MTS

The procedure for making the MPS involved the following five stages:

Stage 1: The PS from Figure 3b, with a volume $V_{\mathrm{PS}}=2\;{\mathrm{cm}}^{3}$, is weighed using an ALN60 type balance (produced by AXIS, Gdansk, Poland). The value obtained is $m_{\mathrm{PS}}=0.050\;g$.

Stage 2: The PS is soaked until saturated with distilled water. The mass of the sponge soaked in distilled water is $m_{\mathrm{PSw}}=0.204\;g$, the mass of water in the sponge is $m_{w}=m_{\mathrm{PSw}}-m_{\mathrm{PS}}=0.154\;g$. The volume occupied by the water is $V_{w}=m_{w}/\rho_{w}=0.154\;\frac{g}{1}g/{\mathrm{cm}}^{3}=0.154{\;\mathrm{cm}}^{3}$. We consider that water replaces the volume of air in the PS. Based on this hypothesis, we find that the volume of air in the PS is $V_{a}=0.154{\;\mathrm{cm}}^{3}$. Under normal conditions of temperature and pressure, the air density is $\rho_{a}=1.29\cdot 10^{-3}\;g/{\mathrm{cm}}^{3}$. Then, the mass of air in the volume of PS is $m_{a}=V_{a}\cdot \rho_{a}≅0.198\cdot 10^{-3}\;g$. The volume of polyurethane fibers in the PS is $V_{f}=V_{\mathrm{PS}}-V_{a}=1.846{\;\mathrm{cm}}^{3}$.

Stage 3: Using the same balance, we weigh 3400 g ($5{\;\mathrm{cm}}^{3}$) of EA and 3153 g ($1{\;\mathrm{cm}}^{3}$) of CI microparticles. The weighed products are mixed in a Berzelius vessel for about 300 s at a temperature of 70 °C ± 5 °C. The temperature, at the surface of the mixture, is measured using an infrared thermometer, Type AX-6520 from AXIOMET (Bielsko-Biała, Poland) (manufactured in Poland and distributed by Transfer Multisort Elektronic S.R.L., Timișoara, Romania) At the end of this stage, a biphasic liquid (BL) is obtained.

Stage 4: In BL, at a temperature of 70 °C ± 5 °C, we insert the PS (from Figure 3b) and continue mixing. After about 200 s, the PS is extracted with absorbed BL and fixed in a vacuumed electric jar (Auto-Vacuum Food System type, made in Taiwan). In the vacuum volume, the excess BL is drained and the EA evaporated. After about 48 h, the MPS is obtained (Figure 5a,b).

Stage 5: Using the same balance, the mass of the MPS is obtained as $m_{\mathrm{MPS}}=0.122\;g$, and the mass of the microparticles in MPS is $m_{\mathrm{CI}}=m_{\mathrm{MPS}}-m_{\mathrm{PS}}=0.072\;g$; it occupied, in the body of MPS, a volume of $V_{\mathrm{CI}}=m_{\mathrm{CI}}/\rho_{\mathrm{CI}}=0.0092{\;\mathrm{cm}}^{3}$. The volume of air in the MPS body is $V_{\mathrm{aMPS}}=V_{\mathrm{PS}}-\left(V_{f}+V_{\mathrm{CI}}\right)=0.1448\;{\mathrm{cm}}^{3}$. The air mass in the MTS is $m_{\mathrm{aMPS}}=V_{\mathrm{aMPS}}\cdot \rho_{a}\approx 0.001\;g$.

The mass and volumetric fractions of the CI microparticles, the polyurethane microfibers, and the air in the MTS body are given in Table 1.

For the calculation of the mass fractions, we refer to the mass $m_{\mathrm{MPS}}$ and to the mass deduced for the CI microparticles, polyurethane microfibers, and air. Instead, for the volume fractions, we refer to the volume of MPS, identical to that of PS, and to the volumes of CI microparticles of polyurethane microfibers, deduced above.

Photographs of the MTS were taken with a zoom-type manual digital microscope (ANDYLUC BESTMAG SRL, China). It is observed (Figure 6a) that the CI microparticles from the MTS are fixed on the polyurethane microfibers. In the presence of the magnetic field (Figure 6b), the CI microparticles are oriented along the magnetic field lines. Highlighting of the columns was performed by supplementing the number of CI microparticles deposited on the MTS.

The MPS has, in its structure C, O, and Fe atoms (Figure 7a). The crystallographic structure of the MPS is shown in Figure 6b. The crystal phase of the MPS was investigated using a PANalytical diffractometer, with Cu-Kα radiation (λ = 0.15406 mm) and 2θ range from 10° to 90°. It is observed from Figure 7b that, in the MPS structure, there is a crystalline phase specific to CI microparticles [47,48] and an amorphous phase characteristic of polyurethane nanofibers [49]. The peak in Figure 7b is characteristic of nanometric crystallites, of type α-Fe, for CI microparticles that are not chemically treated [50].

It is known that the saturation magnetization of composites is dependent on the volume fraction of CI microparticles [51,52]. Based on this conclusion, the Equation $\mu_{0}\sigma_{\mathrm{SMTS}}=\varphi_{\mathrm{CI}}\mu_{0}\sigma_{\mathrm{SCI}}$ can be obtained (where $\mu_{0}$ is the vacuum permeability, $\sigma_{\mathrm{SMTS}}$ is the relative saturation magnetization of MPS, $\varphi_{\mathrm{CI}}$ is the volume fraction of the CI microparticles in MPS, and $\sigma_{\mathrm{SCI}}$ is the relative saturation magnetization of the microparticles CI) [53], and by using the magnetization curve from Figure 2, we obtain the magnetization curve of the MPS, as shown in Figure 8.

It can be seen from Figure 8 that the MPS magnetization slope has the same shape as that of the CI microparticles. However, due to the volume fraction of the CI microparticles, i.e., value $\varphi_{CI}=0.46\;\mathrm{vol}\%$, the saturation magnetization of the MTS sponge is 266 times lower compared with that of the CI microparticles from Figure 2.

The adhesive part of the CS is applied on the faces of the MPS by rolling. At the end of this step, the MTS device from Figure 9 is obtained.

It can be seen from Figure 9 that the MTS device has a stable mechanical configuration and is provided with copper wires for electrical connections. The copper foil with a thickness of 0.05 mm is molded on the rough surface of the MTS sponge, which explains the unevenness, shown in Figure 9, on the faces of the MTS.

### 2.3. Experimental Set-Up and Measurements

The experimental set-up for the study of the MTS in a magnetic field superimposed with the action of the mechanical force field is shown in Figure 10.

The set-up includes a direct current electromagnet, consisting of a magnetic core (Figure 10, (1)) and a coil (Figure 10, (2)) connected to the direct current source (type RXN-3020D, from Electronics Co., Ltd., Nagoya, Japan). The MTS is fixed by the axis (Figure 10, (3)) between the magnetic poles N and S. By adjusting the intensity of the direct current discharged by the DCS source through the electromagnet coil, the H value of the intensity of the incident magnetic field is adjusted at the MTS and measured by the Hall probe (h) connected to the gaussmeter (Gs) (type DX-102, from DexingMagnet, Xiamen, China). The MTS sensor is inserted between two stratisticlotextolite plates, which, for simplicity, are not represented in Figure 10. The mechanical resistance of these plates favors the uniform application of the force F, exerted by the weight of the non-magnetic masses (5), on the MTS through the axis (3). The marked mass (m) exerts compression pressure (mechanical stress) (*p*) (Figure 11) on the MTS without affecting the Hall probe.

The RLC bridge (BR) (model 8846A, Fluke, Everett, WA, USA) connected to the MTS is used to measure, in time and for determined durations, the electrical capacity depending on the force, F, in the absence and in the presence of the magnetic field.

## 3. Results and Discussion

The RLC bridge is set on the electrical capacity measurement range, C, at time intervals ∆t = 1 s.

The C values of the electrical capacity of MTS as a function of time, t, over a period of 180 s, in the presence of the compression force, F, but in the absence of the magnetic field, are plotted in Figure 12a.

It can be seen from Figure 12a that the electrical capacity values, C, are constant in time but vary with increasing F. It is observed that, for an increase in the force F to a value of F + ∆F, the quantities C increase to values of C + ∆C, where ∆C = 0.001 nF, for ∆F = 1 N. When repeating the measurements, after five cycles of five recorders, the deviations were within ±2%, identical to the measurement error of the RLC bridge. When applying forces F higher than 5 N, the electrical capacity, C, of the MTS remains constant in time. For m≥650 g, the values of C do not return to the initial values when the deformation is removed.

In order to explain the behavior of MTS subjected to external magnetic and deformation fields, in the Appendix A, we propose a model of MTS for three different situations: A. MTS subjected to a field of compression forces; B. MTS subjected to a magnetic field; C. MTS subjected to a magnetic field superimposed on a field of compression forces.

In Equation (4) from the Appendix A, we introduce the quantities $C_{m0}$ and $C_{m}$ using the function $C_{m}=C_{m}\left(F\right)$ from Figure 12b, and we obtain $\varepsilon_{\mathrm{zz}}=\varepsilon_{\mathrm{zz}}\left(F\right)$, as shown in Figure 12c. Figure 12c shows that the deformation of MPS has a linear dependence on F, and the modulus of the deformation components increases with an increase in F.

Using the model described at point A from the Appendix A, with $h_{0}=0.4\;\mathrm{mm}$, and with the data from Figure 12b,c, we obtain the elasticity constant value k≈12.26 kN/m and the Young’s modulus E=10 kPa of MPS in the absence of the magnetic field.

Next, for the fixed quantities F and H, we measure, at intervals ∆t = 1 s for a duration of 180 s, the $C_{H}$ values of the electrical capacity of the same MTS. The obtained values are plotted in Figure 13.

It can be observed from Figure 13 that, on the one hand, the quantities $C_{H}$ for fixed F and H values are constant in time. On the other hand, the $C_{H}$ values for constant F quantities increase with the increasing H intensity of the magnetic field. To appreciate the variation with time, t, of the electrical capacity, we calculate the average value of the functions $C_{H}=C{\left(t\right)}_{F,\;H}$ from Figure 13, and we obtain the functions $C_{\mathrm{Hm}}=C_{\mathrm{Hm}}{\left(H\right)}_{F}$, as shown in Figure 14a.

It can be seen from Figure 14a that the functions $C_{\mathrm{Hm}}=C_{\mathrm{Hm}}{\left(H\right)}_{F}$ can be approximated by:(1)$$C_{\mathrm{Hm}}=C_{{\mathrm{Hm}}_{0}}+\alpha_{C}\cdot H-\beta_{C}\cdot H^{2}$$
where $C_{{\mathrm{Hm}}_{0}}$, $\alpha_{C}$, and $\beta_{C}$ have the values shown in Table 2.

Using the model of the dipolar magnetic approximation (Figure A2 of the Appendix A) from Equation (A32), it is observed that the electric capacity of MTS, in the absence of F, increases significantly with the increase in the magnetic field, H, in agreement with the data obtained in Figure 14a for F = 0 N. On the other hand, when applying F and H, according to the model from paragraph B of the Appendix A given by Equation (A36), the effects are cumulative, in accordance with the corresponding experimental data from Figure 14a.

The components $e_{\mathrm{zzH}}$ of the deformations of MPS located in the field of deforming forces superimposed over a magnetic field are defined by the Equation (A39):(2)$$e_{zzH}=\frac{h_{m}}{h_{0}}-1=\frac{C_{Hm_{0}}}{C_{Hm}}-1$$
where $h_{m}$ is the thickness of MPS in MTS, with the electrical capacity m, at values H ≠ 0 and F ≠ 0, and $h_{0}$ is the thickness of MPS in MTS, with the electrical capacity $C_{{\mathrm{Hm}}_{0}}$, at values H ≠ 0 and F = 0.

In Equation (2), we introduce the values of $C_{{\mathrm{Hm}}_{0}}$ and $C_{\mathrm{Hm}}$ corresponding to the functions $C_{\mathrm{Hm}}=C_{\mathrm{Hm}}{\left(H\right)}_{F}$, from Figure 14a, and we obtain the functions $e_{\mathrm{zzH}}=e_{\mathrm{zzH}}{\left(H\right)}_{F}$, represented in Figure 14b. It is observed from Figure 14b that, in the magnetic field superimposed over the field of mechanical forces, the allure of the functions $e_{\mathrm{zzH}}=e_{\mathrm{zzH}}{\left(H\right)}_{F}$ is that of a second-order polynomial function and not that of a linear polynomial function obtained in Figure 12c for MTS subjected only to the action of F. From the same figure, it is observed that the modulus of the quantities $e_{\mathrm{zzH}}$ increases with an increase in the quantity H, at values F = const. Likewise, for the same H, the modulus of the values of the deformation components increases with the increasing size of F.

The contribution $\alpha_{F}$ brought by the force F to an increase in the electrical capacity of MTS, in the absence of the magnetic field, can be obtained from the equation:(3)$$\alpha_{F}\left(\%\right)=\left(\frac{C_{m}}{C_{m0}}-1\right)100$$
where $C_{m}$ and $C_{m0}$ are the electric capacities of the MTS for F≠0 and F=0, respectively.

If, in Equation (3), we introduce the function $C_{m}=C_{m}\left(F\right)$ from Figure 12b, we obtain $\alpha_{m}=\alpha_{m}\left(F\right)$, represented in Figure 15a.

It is observed from Figure 15a that $\alpha_{F}$ increases linearly with F values. For F = 5 N, we obtain $\alpha_{F}=11.11\%$.

For the MTS located in the magnetic field, H, superimposed over the field of deforming forces, F, we denote by $\alpha_{H}$ the contribution brought by the magnetic field to an increase in the value of the electrical capacity of MTS, which can be expressed by the equation:(4)$$\alpha_{H}\left(\%\right)=\left(\frac{C_{\mathrm{Hm}}}{C_{m_{0}}}-1\right)100$$
where $C_{\mathrm{Hm}}$ is the electrical capacity of MTS for H≠0 and F≠0, and $C_{m_{0}}$ is the electrical capacity of MTS in the absence of both magnetic and deformation fields.

In Equation (4), we introduce the functions $C_{\mathrm{Hm}}=C_{\mathrm{Hm}}{\left(H\right)}_{F}$ from Figure 14a, and we obtain the function $\alpha_{H}=\alpha_{H}{\left(H\right)}_{F}$, as in Figure 15b.

In the case of the absence of the magnetic field (Figure 15a), by applying the force F, the contribution brought to an increase in the MTS electrical capacity is:(5)$$\alpha_{F}=\left\{\begin{array}{c} 2.22\%\;\mathrm{for}\;F=1N\\ 4.44\%\;\mathrm{for}\;F=2N\\ 6.66\%\;\mathrm{for}\;F=3N \end{array}\right.$$

When applying the magnetic field in the absence of deforming forces (Figure 15b), the contribution brought to an increase in the electrical capacity of MTS is 73.21% at H = 280 kA/(m). Then, when applying the deforming forces superimposed over the magnetic field of intensity H = 280 kA/m, the contribution brought by H to an increase in $C_{\mathrm{Hm}}$ is:(6)$$\alpha_{H}=\left\{\begin{array}{c} 100,64\%\;\mathrm{for}\;F=1N\\ 157,46\%\;\mathrm{for}\;F=2N\\ 224,15\%\;\mathrm{for}\;F=3N \end{array}\right.$$

From the results (5) and (6), it is observed that, by applying the magnetic field, the sensitivity of MTS to the action of the deforming forces increases significantly.

Using Equations (A39)–(A42) from the Appendix A in correlation with the magnitudes of electrical capacities from Figure 14a, we obtain the function $k_{\mu F}=k_{\mu F}\left(H\right),$ as shown in Figure 16a, and the function $E_{\mathrm{HF}}=E_{\mathrm{HF}}\left(H\right),$ as shown in Figure 16b, when MTS is subjected to a magnetic field superimposed on a field of compression forces.

It can be seen from Figure 16a,b that the effects generated by the magnetic field, H, and the deformation forces, F, overlap. The dependence of the two elasticity constants on the magnetic field is due to magnetoconstriction. The effect is specific to materials containing magnetizable microparticles [42,43,44,45,51,54].

## 4. Conclusions

The magneto-tactile sensor (MTS) is fabricated by using a commercially used polyurethane sponge (Figure 3), ethanol, carbonyl iron microparticles, and copper foil with electroconductive self-adhesive (Figure 9). With the installation in Figure 10, the electrical capacity of the MTS is measured in time and for determined durations. The values of the electric capacity do not change in the time of measurements and increase linearly with an increase in the deforming forces (Figure 12) in the absence of a magnetic field. When applying a magnetic field, and in the absence of mechanical deformations, the electrical capacity of the MTS (Figure 14a,b) increases significantly with increasing magnetic field strength. By superimposing the magnetic field over the field of deformations, consisting of forces up to 3 N, (Figure 14a), a significant increase in the electrical response provided by the MTS is observed, (Figure 15b). Therefore, by applying a magnetic field, it is possible to achieve a controlled increase in the MTS sensitivity.

The elasticity constant (Figure 16a) and Young’s modulus (Figure 16b) are strongly influenced by the external magnetic field. The proposed theoretical model (see Appendix A) qualitatively describes the mechanisms leading to the experimental results obtained in this paper.

## Figures and Tables

**Figure 1 nanomaterials-12-03231-f001:**
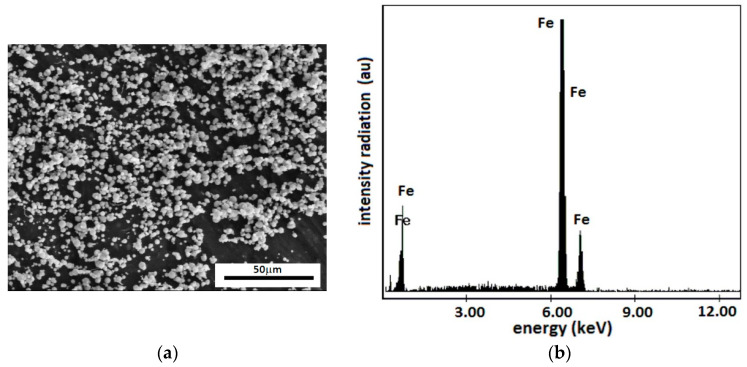
(**a**) SEM morphology of CI microparticles; (**b**) EDX spectra for the elemental analysis of CI microparticles.

**Figure 2 nanomaterials-12-03231-f002:**
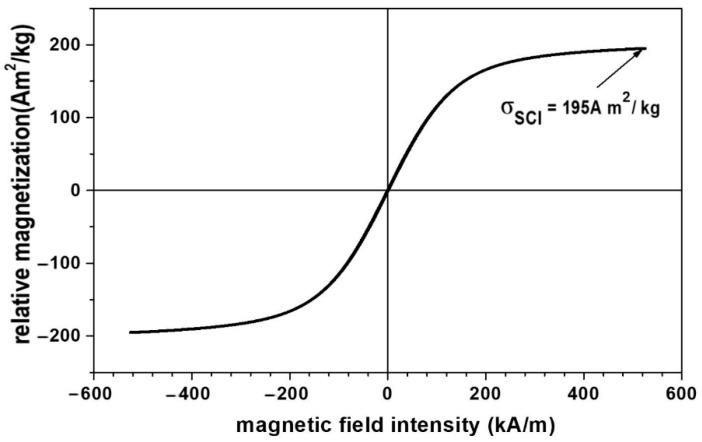
Magnetization slope for the CI microparticles.

**Figure 3 nanomaterials-12-03231-f003:**
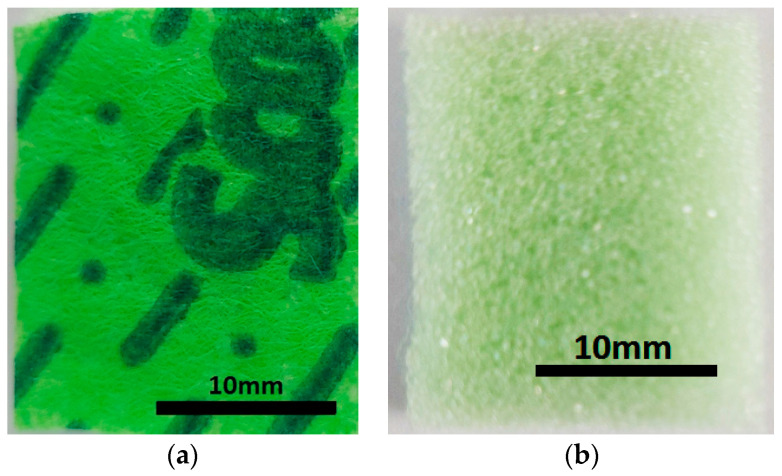
Photo images: (**a**) Super absorbent cloth (AC); (**b**) polyurethane sponge (PS).

**Figure 4 nanomaterials-12-03231-f004:**
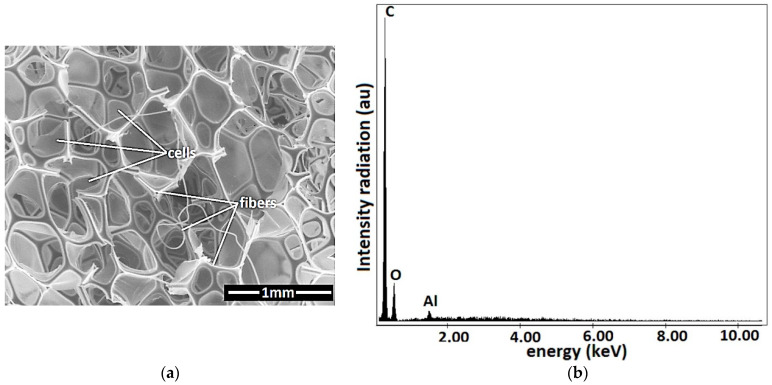
(**a**) SEM morphology of PS; (**b**) EDX spectra for the elemental analysis of PS.

**Figure 5 nanomaterials-12-03231-f005:**
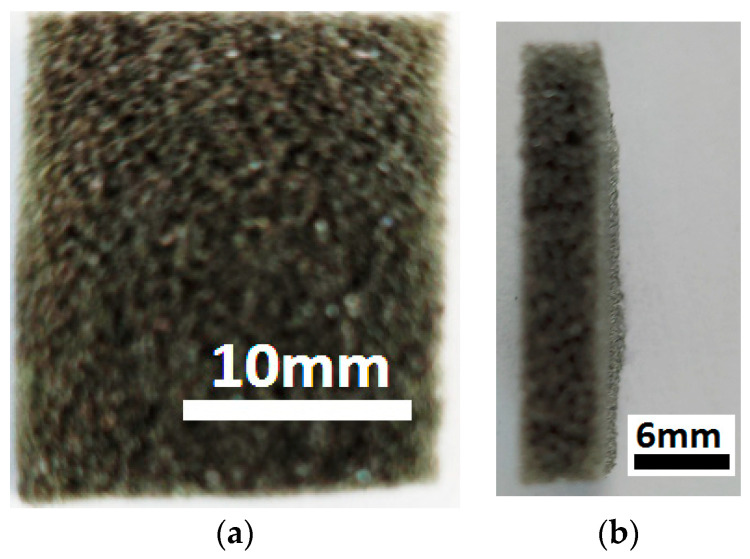
MPS photos: (**a**) Front view; (**b**) side view.

**Figure 6 nanomaterials-12-03231-f006:**
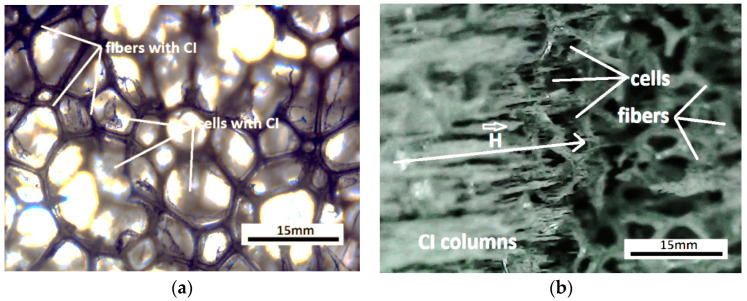
The MPS viewed under a digital microscope: (**a**) In the absence of the magnetic field; (**b**) in the presence of the magnetic field (H≈40kA/m).

**Figure 7 nanomaterials-12-03231-f007:**
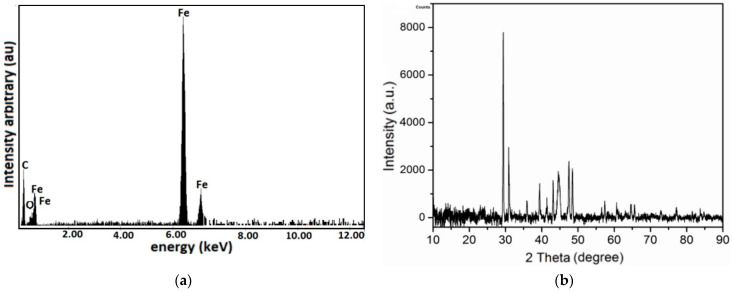
(**a**) EDX spectra for the elemental analysis of MPS; (**b**) XRD analysis of MPS.

**Figure 8 nanomaterials-12-03231-f008:**
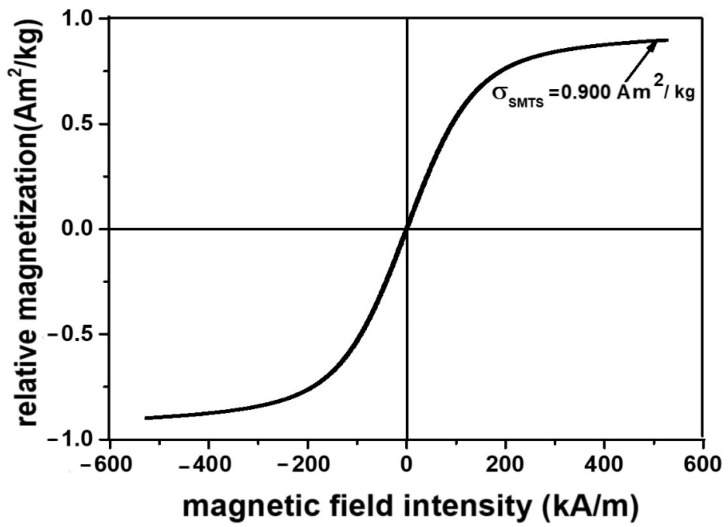
Magnetization slope for the MPS.

**Figure 9 nanomaterials-12-03231-f009:**
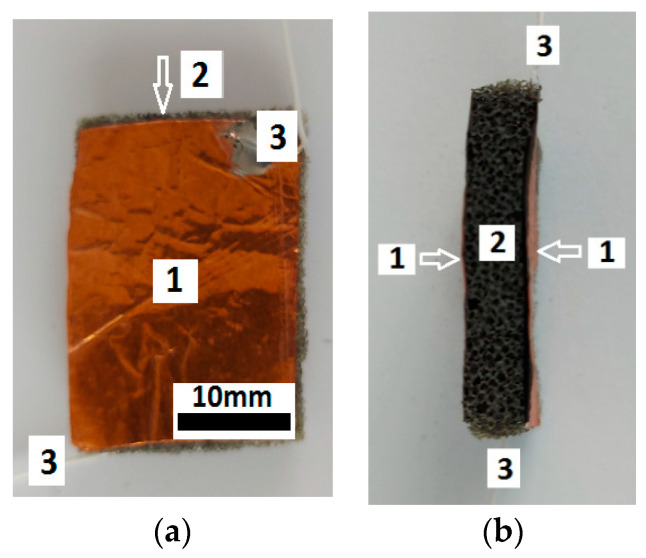
(**a**) Front view photo of the MTS; (**b**) lateral view photo of the MTS (1, copper foil; 2, MTS; 3, electrical connection).

**Figure 10 nanomaterials-12-03231-f010:**
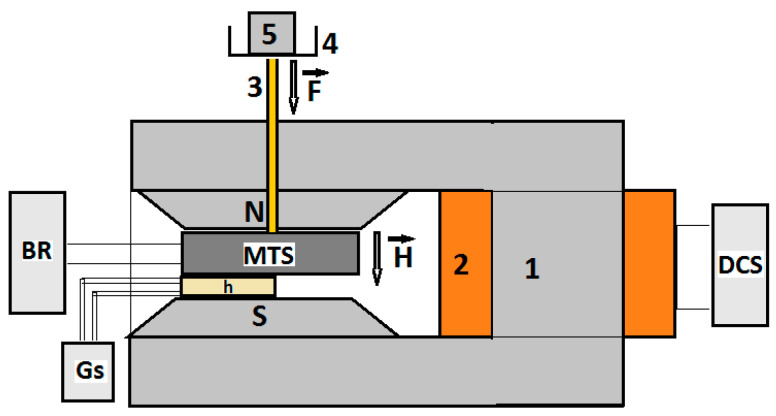
Experimental set-up (overall configuration): (1) Magnetic yoke; (2) coil; (3) non-magnetic axis; (4) plate; (5) marked mass made of lead. BR, RLC bridge; DCS, direct current source; Gs, Gaussmeter; N and S, magnetic poles; MTS, magneto-tactile sensor; h, Hall probe; $\vec{F}$, compression force vector; $\vec{H}$, magnetic field strength vector.

**Figure 11 nanomaterials-12-03231-f011:**
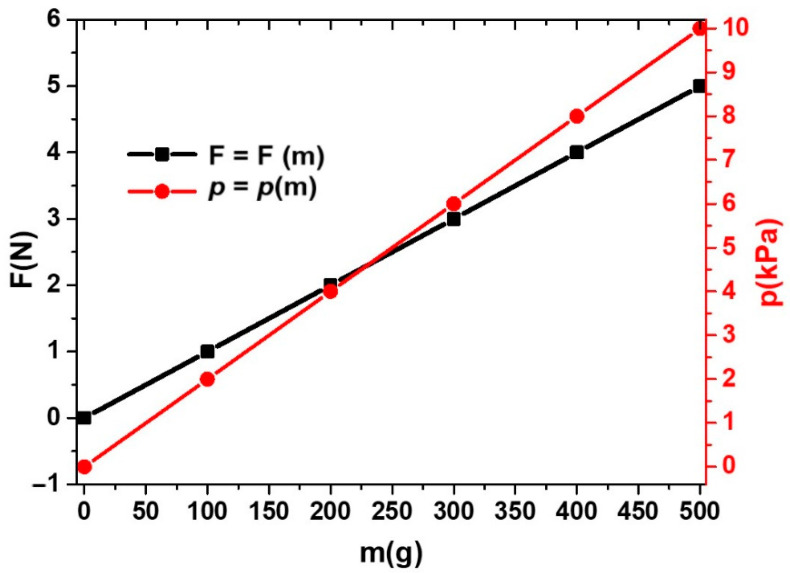
Compression force (F) and compression pressure (*p*) induced on the MTS by the marked masses (m).

**Figure 12 nanomaterials-12-03231-f012:**
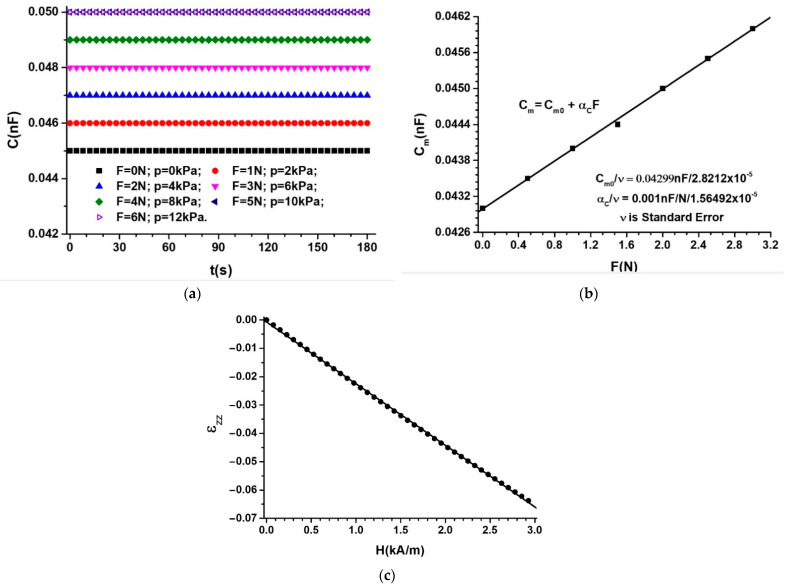
(**a**) The electrical capacity, C, of MTS, as a function of time, t, with the force F as a parameter; (**b**) the average capacity $C_{m}$ of MTS as a function of F; (**c**) the components $\varepsilon_{\mathrm{zz}}$ of the deformations of MPS as a function of F (dots are experimental data and lines are the first-order polynomial fit).

**Figure 13 nanomaterials-12-03231-f013:**
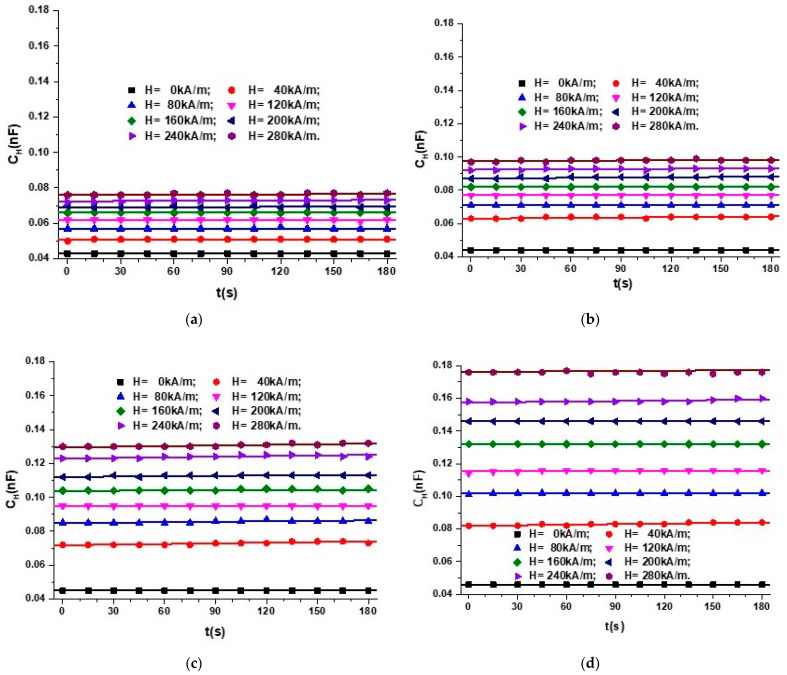
The electrical capacity of MTS, $C_{H}$, as a function of time and intensity of the magnetic field applied on the direction of the deformation force: (**a**) F = 0 N; (**b**) F = 1 N; (**c**) F = 2 N; (**d**) F = 3 N.

**Figure 14 nanomaterials-12-03231-f014:**
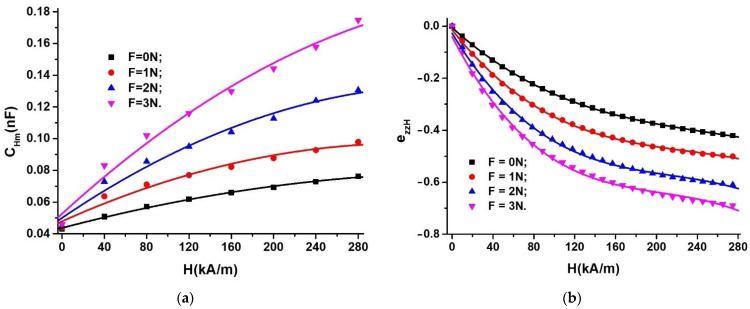
(**a**) The average electrical capacity, $C_{\mathrm{Hm}}$, depending on the intensity of the magnetic field, H, for the force, F, as a parameter; (**b**) the components $e_{\mathrm{zzH}}$ depending on the intensity of the magnetic field, H, for the force, F, as a parameter (dots are experimental data and lines are the second-order polynomial fit).

**Figure 15 nanomaterials-12-03231-f015:**
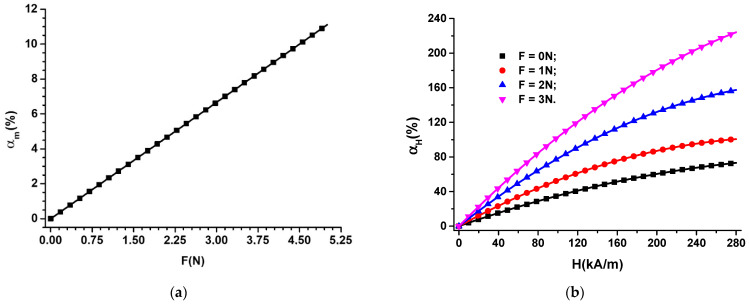
(**a**) The variation of $\alpha_{F}$ as a function of the deformation force, F; (**b**) the contribution $\alpha_{H}$ as a function of the magnetic field intensity, H, for the deforming force, F, as a parameter.

**Figure 16 nanomaterials-12-03231-f016:**
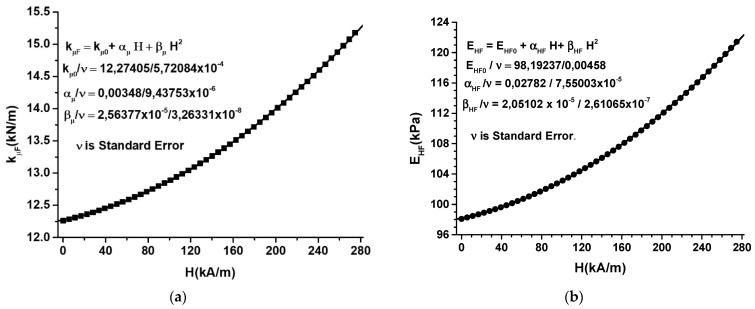
(**a**) The variation of the elastic constant of MPS, $k_{\mu F}$, as a function of the magnetic field intensity, H; (**b**) the variation of Young’s modulus, $E_{\mathrm{HF}}$, as a function of the magnetic field intensity, H.

**Table 1 nanomaterials-12-03231-t001:** Mass (Φ) and volumetric (φ) fractions for $MS_{i}$, where i=1, 2, and 3 denote sample numbers.

$\Phi_{CI}\left(\%wt.\right)$	$\Phi_{f}\left(\%wt.\right)$	$\Phi_{a}\left(\%wt.\right)$	$\varphi_{CI}\left(\%vol.\right)$	$\varphi_{f}\left(\%vol.\right)$	$\varphi_{a}\left(v\%vol.\right)$
59	40	1	0.46	92.3	7.24

**Table 2 nanomaterials-12-03231-t002:** Values of $C_{{\mathrm{Hm}}_{0}}$, $\alpha_{C}$, and $\beta_{C}\;$ for F=0÷3N.

$F\left(N\right)$	$C_{Hm_{0}}\left(nF\right)/\nu$	$\alpha_{C}\left(nF\cdot m/kA\right)/\nu$	$\beta_{C}\left(nF\cdot m^{2}/kA^{2}\right)/\nu$
0	$0.04367/5.86282\cdot 10^{-4}$	$1.75845\cdot 10^{-4}/9.78162\cdot 10^{-6}$	$2.20238\cdot 10^{-7}/3.359092\cdot 10^{-8}$
1	0.04786/0.00271	$2.99281\cdot 10^{-4}/4,51904\cdot 10^{-5}$	$4.54501\cdot 10^{-7}/1.55184\cdot 10^{-7}$
2	0.05001/0.0036	$4.52734\cdot 10^{-4}/6.00749\cdot 10^{-5}$	$6.12388\cdot 10^{-7}/2.06298\cdot 10^{-7}$
3	0.05264/0.0048	$6.09759\cdot 10^{-4}/8.00487\cdot 10^{-5}$	$6.72545\cdot 10^{-7}/2.74888\cdot 10^{-7}$

Here, ν is the standard error.

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
