# Peer review of "Magneto-Tactile Sensor Based on a Commercial Polyurethane Sponge"

_nanomaterials, 2022, doi:10.3390/nano12183231_

Round 1

Reviewer 1 Report

Article “Magneto-tactile sensor based on commercial polyurethane sponge” describes the creation add the study of a magneto-tactile sensor (MTS) based on a polyurethane sponge doped with microparticles of carbonyl iron. Measured capacitances are dependent on the applied force and external magnetic field. When acting together, over 220% increase in capacitance can be detected. The article is written carefully but can be slightly improved. Some text corrections and vague statements are mentioned below. Please answer the questions in the main text.

Is the marked mass applied at one point or on the entire sponge surface? Is the force filed in the sponge homogeneous?

What is the maximum mass after which the foam does not return to its initial volume, capacitance etc..?

Authors should consider doing the same experiment in an inhomogeneous magnetic field. I am curious about the result (but that could be a different article).

Line 167 is not  “…magnetic count of the vacuum”  but  Vacuum permeability

Figure 11 description of the vertical axis missing

Figure 12  a) 180 points should be in the figure, and is 7?

Figure 12 b) Please provide an algebraic formula in the plot or text for the dependence C(F); almost there in line 222, but go further and write the equation.

Line 222 “From the comparison of the electrical capacity values, obtained in Figure 12a,b, it is observed that the instantaneous values coincide with the average values.” I cannot follow. You did not show the instabilities which appear directly after pressing. The black line (Fig 12a) changes after applying 10kPa to the blue line with a delay and small damped oscillation (that is my assumption).    

Line 337 Can you explicitly write the functions C(H,F). I hope the weights used to create pressure were not metallic and were not additionally pulled by the magnetic field when switched on.

Equation 41   Ideal case for a 2D plot C=f(F,H), authors should think about.

Line 456 The values of the electric capacity do not change in time and increase linearly with an increase in the deforming forces (Figure 12) in the absence of a magnetic field. They do not change in time of measurement, not in general

Reviewer 2 Report

This manuscript reports a magneto-tactile sensor (MTS) fabricated with polyurethane sponge and carbonyl iron particles, supporting by materials characterizations, and performance tests. The paper is completed in the present form, but it is a bit lengthy. Some results can be attached as supporting materials.

Reviewer 3 Report

In this paper (1853722), a magneto-tactile sensor based on a polyurethane sponge is presented. The results are acceptable and the topic can attract a relatively wide range of readerships. But there are many problems in the introduction, presentation, and discussion of the results. As such, a Major revision is needed before possible publication. My specific comments are as follows:

1.      please further clarify the differences between this work and your recently published work, Polymers 2022, 14(10), 2062.

2.      The logic, motivation, and innovation of the manuscript are unclear, including unclear motivation for material selection nor sensor development, and insufficient article researches reviewed about tactile sensors based on polyurethane sponge.

3.      Please provide mechanical properties of the sensor, such as Young’s modulus, tensile and compression properties.

4.      Capacitance test: The authors should provide the effect of the test frequency on the sensor performance.

5.      This paper aims to develop tactile sensors. However, some key performance parameters of the sensor have not been studied, such as dynamic response curves of the sensor under different pressures, response/recovery times, repeatability and hysteresis.

6.      Abbreviations are not used properly. The abbreviation should be given by following its full spelling for the first time. For example, CI.

7.      Figures’ quality needs to be improved. For example, the vertical axis of Figure 11 lacks a title. It’s better to use double Y-axis to

8.      Most of the references are out of date. It is suggested that references should be concentrated in the last three years.

9.      English writing of the manuscript needs further polishing.

Round 2

Reviewer 1 Report

The authors corrected the main text and answered the raised questions. The articles have improved well. I have no further questions. Although the authors should improve the Figures' quality – axes labels are almost unreadable, e.g. Fig 1b, 2, 4b 7ab, 8, 11ab, 12, 13, 14, 16.

Reviewer 3 Report

All the comments are answered appropriately. Thus this manuscript could be accepted.

Author Response

Dear Reviewer 3,

Thank you for reviewing our manuscript!

Best regards,

The Authors